# Incipient Parallel Evolution of SARS-CoV-2 Deltacron Variant in South Brazil

**DOI:** 10.3390/vaccines11020212

**Published:** 2023-01-18

**Authors:** Fernando Hayashi Sant’Anna, Tiago Finger Andreis, Richard Steiner Salvato, Ana Paula Muterle Varela, Juliana Comerlato, Tatiana Schäffer Gregianini, Regina Bones Barcellos, Fernanda Marques de Souza Godinho, Paola Cristina Resende, Gabriel da Luz Wallau, Thaís Regina y Castro, Bruna Campestrini Casarin, Andressa de Almeida Vieira, Alexandre Vargas Schwarzbold, Priscila de Arruda Trindade, Gabriela Luchiari Tumioto Giannini, Luana Freese, Giovana Bristot, Carolina Serpa Brasil, Bruna de Oliveira Rocha, Paloma Bortolini Martins, Francine Hehn de Oliveira, Cock van Oosterhout, Eliana Wendland

**Affiliations:** 1Hospital Moinhos de Vento, Porto Alegre 90035-000, RS, Brazil; 2Centro de Desenvolvimento Científico e Tecnológico, Centro Estadual de Vigilância em Saúde, Secretaria Estadual da Saúde do Rio Grande do Sul (CDCT/CEVS/SES-RS), Porto Alegre 90450-190, RS, Brazil; 3Laboratório Central de Saúde Pública, Centro Estadual de Vigilância em Saúde, Secretaria Estadual da Saúde do Rio Grande do Sul (LACEN/CEVS/SES-RS), Porto Alegre 90450-190, RS, Brazil; 4Laboratory of Respiratory Viruses and Measles, Oswaldo Cruz Institute (IOC), Oswaldo Cruz Foundation (FIOCRUZ), Rio de Janeiro 21040-900, RJ, Brazil; 5Departamento de Entomologia e Núcleo de Bioinformática, Instituto Aggeu Magalhães, Fundação Oswaldo Cruz Pernambuco (FIOCRUZ-PE), Recife 50740-465, PE, Brazil; 6Departamento de Análises Clínicas, Universidade Federal de Santa Maria, Santa Maria 97105-900, RS, Brazil; 7School of Environmental Sciences, University of East Anglia, Norwich Research Park, Norwich NR4 7TJ, UK; 8Graduate Program in Biosciences, Federal University of Health Sciences of Porto Alegre (UFCSPA), Porto Alegre 90050-170, RS, Brazil

**Keywords:** COVID-19, SARS-CoV-2 genomes, gene flow, recombination, genetic introgression, adaptive landscape, severe acute respiratory syndrome coronavirus 2, Brazil, Deltacron, recombinant, AYBA-RS

## Abstract

With the coexistence of multiple lineages and increased international travel, recombination and gene flow are likely to become increasingly important in the adaptive evolution of SARS-CoV-2. These processes could result in genetic introgression and the incipient parallel evolution of multiple recombinant lineages. However, identifying recombinant lineages is challenging, and the true extent of recombinant evolution in SARS-CoV-2 may be underestimated. This study describes the first SARS-CoV-2 Deltacron recombinant case identified in Brazil. We demonstrate that the recombination breakpoint is at the beginning of the Spike gene. The 5′ genome portion (circa 22 kb) resembles the AY.101 (Delta), and the 3′ genome portion (circa 8 kb nucleotides) is most similar to the BA.1.1 (Omicron). Furthermore, evolutionary genomic analyses indicate that the new strain emerged after a single recombination event between lineages of diverse geographical locations in December 2021 in South Brazil. This Deltacron, AYBA-RS, is one of the dozens of recombinants described in 2022. The submission of only four sequences in the GISAID database suggests that this lineage had a minor epidemiological impact. However, the recent emergence of this and other Deltacron recombinant lineages (XD, XF, and XS) suggests that gene flow and recombination may play an increasingly important role in the COVID-19 pandemic. We explain the evolutionary and population genetic theory that supports this assertion, concluding that this stresses the need for continued genomic surveillance. This monitoring is vital for countries where multiple variants are present, as well as for countries that receive significant inbound international travel.

## 1. Introduction

During the first two years of the COVID-19 outbreak, most genetic variation in SARS-CoV-2 was generated by mutations, some of which improved the fitness of the virus in its new host and epidemiological environment. More recently, the adaptive evolution of SARS-CoV-2 has also involved recombination. In SARS-CoV-2, recombination can occur when distinct variants co-infect the same host cell and exchange genetic material [1,2]. This process is called genetic introgression, and it plays an essential role in the virulence evolution of parasites and pathogens [3]. Gene flow occurs when a genotype of a given variant is moved from one population into another. If this gene flow also results in a co-infection in a host already infected with another variant, it might lead to genetic introgression. This phenomenon is known to have resulted in the evolution of novel subspecies in other human parasites, such as *Cryptosporidium* spp. (e.g., [4]). The exchange of genetic information between distinct lineages underpins the virulence evolution of these parasites [5]. In the case of SARS-CoV-2, international travel is likely to contribute considerably to the gene flow of different variants across the globe, thereby increasing the probability of genetic introgression [3].

Genetic introgression offers three potential advantages over mutation: (1) it can insert multiple substitutions all at once; (2) these substitutions have been previously selected and are functional in the genomic background of the parental lineage; and (3) this enables the recombinant genotype to bridge the fitness valleys in the adaptive fitness landscape and find higher fitness peaks [6]. In addition, increased international travel enables gene flow and recombinant exchange between distinct lineages that have evolved worldwide. Consequently, recombination and gene flow may play an increasingly important role in the transmissibility, severity, and resistance to vaccines and treatments of SARS-CoV-2, and the evolutionary epidemiology of the COVID-19 pandemic. Here, we examine the evidence for the incipient parallel evolution of recombinant lineages, studying SARS-CoV-2 genomes in Brazil. This country has seen less intensive genomic and epidemiological surveillance than other parts of the world. Hence, by studying the SARS-CoV-2 genome sequence variation in Brazil, we may better understand the extent of cryptic recombinant lineages.

Up to September 2022, the WHO reported five SARS-CoV-2 variants of concern (VOCs): Alpha, Beta, Gamma, Delta, and Omicron [7]. The variant Delta (B.1.617.2) emerged in India at the end of 2020 and spread to at least 185 nations [8,9]. The WHO then classified this variant as a VOC due to its high transmissibility and potential to cause severe COVID-19. In November 2021, the Omicron (BA.1) variant emerged in South Africa [10] and was declared a VOC. According to GISAID in August 2022, the sub-variants of BA.1 spread to at least 193 countries [11,12]. The widespread and simultaneous circulation of both VOCs, Omicron and Delta, resulted in recombinants known as “Deltacron”. Genomic analysis of SARS-CoV-2 samples revealed a novel Deltacron lineage in France in February 2022. This lineage presented two recombination breakpoints, one at the beginning of the Spike region and another at the beginning of ORF3a [13]. The genomic segment within these limits displayed Omicron signature mutations; however, the rest of the genome presented Delta signature mutations. This recombinant variant was then designated XD [13] and was mainly found in Denmark and the Netherlands [14,15]. Besides XD, two other Omicron and Delta hybrids, XF and XS, circulated in the United Kingdom and the USA, respectively. Both recombinants presented a minor Delta portion at the 5’ end of an Omicron genomic backbone, although with distinct breakpoint locations [16]. According to the Cov-Lineages [17], there are less than 40 sequences to each XD, XF and XS recombinant.

Most recombinants in the GISAD database were recovered from European countries and the USA. To fill the gap of studies in other regions, we investigated four putative Brazilian recombinants recovered from South and Southeast Brazil. We analysed the mutation profile, identified the recombination breakpoints, and made a phylogenetic reconstruction to trace the origins of the novel recombinant lineages. This analysis confirmed that the Deltacron recombinant in the country had evolved de novo, and that it could be considered a case of incipient parallel evolution. In other words, this recombinant variant acquired similar characteristics to those of other Deltacron variants (i.e., it shows a high sequence similarity), and the new variant AYBA-RS had evolved this independently from other circulating variants. Our results highlighted the importance of genomic surveillance for monitoring the viral evolution caused by co-infections with different SARS-CoV-2 lineages and for identifying putative recombinants. This action is specifically pressing during periods of high viral circulation and in countries with multiple variants, as well as in regions that are a hub for international air travel.

## 2. Materials and Methods

### 2.1. Bioethics, Sample Collection and Processing

The clinical samples were retrieved from three different institutions performing COVID-19 diagnosis and SARS-CoV-2 genomic surveillance in Rio Grande do Sul, Brazil: Centro Estadual de Vigilância em Saúde-CEVS (The State Centre for Health Surveillance of the State Department of Health); the Genetics and Molecular Biology Laboratory from Hospital Moinhos de Vento; and Laboratório de Bioinformática Aplicada a Microbiologia Clínica from Federal University of Santa Maria (UFSM). In all cases, the SARS-CoV-2 infection was first detected by real-time RT-PCR, and the samples were submitted to a genomic sequencing routine in each institution.

### 2.2. Whole-Genome Sequencing, Assembly, and Quality Control

We observed an S gene dropout (i.e., gene not detected) in the sample SC2-9898 on May 2022 and then selected this sample for genome sequencing with the SARS-CoV-2 FLEX NGS panel (Paragon Genomics, Fremont, CA, USA) on the Illumina MiSeq platform. The library preparation was conducted according to the manufacturer’s protocol, and the sequencing was performed using the MiSeq Reagent Micro Kit v2 (Illumina Inc, San Diego, CA, USA). The FASTq files were obtained using the Local Run Manager Generate FASTQ Analysis Module v3.0 (Illumina Inc, San Diego, CA, USA) and submitted to the SOPHiA DDM v.5 platform. These files were analysed using the CleanPlex SARS hCoV2 pipeline for sequence alignment. Finally, the sequence was deposited in the GISAID database with the entry EPI_ISL_14381991.

For the EPI_ISL_12110384 and EPI_ISL_14284846 sequences, whole-genome sequencing was performed using the Illumina COVIDSeq protocol (Illumina Inc,, San Diego, CA, USA) on the Illumina sequencing platform. The pipeline ViralFlow was used to perform genome assembly, variant calling, and consensus generation [18]. To evaluate the quality and determine the lineage of the genome sequences, we analysed them on Nextclade Web version 2.3.1 [19].

### 2.3. Identification of Lineage Counterparts

To identify the candidate parental genomes that may have introgressed to from the new Brazilian recombinant, we performed blast searches using the sequence Brazil/RS-FIOCRUZ-8390/2022 (EPI_ISL_12110384) on the “Unassigned” dataset from GISAID (assessed on 25 July 2022). After this, we visually inspected the mutation pattern of the top hits on the Nextclade Web, using the putative Brazilian recombinant sequences as references.

### 2.4. Parental Lineages Determination

Genome sequences were first aligned using Nextalign version 1.11.0 with default parameters and sequence MN908947 as reference. We evaluated the recombinant genomes using Sc2rf [20]. Subsequently, we manually segmented the genome of the oldest sequences of each recombinant lineage according to the Delta and Omicron portions using Aliview version 1.27 [21]. Next, we assessed the Pangolin lineage of each of the 5′ delta and the 3′ omicron segments in the Nextclade Web and Pangolin COVID-19 Lineage Assigner version 4.1.1 [22].

We also performed a blastn search (BLAST version 2.10.1+, [23,24]) for each of the Delta and Omicron segments in the sequences of Nextstrain’s global analysis—GISAID data [25] (assessed on 23 June 2022). We then checked the lineage of the oldest top-hit strain in the GISAID metadata.

We built lineage-specific databases (GISAID sequences) considering the Pango lineages determined in the previous analyses. We again utilised each segment as a query to find the top 20 best hits of the reference databases (Delta or Omicron). Once we had identified the best parental candidates, we downloaded their sequences from GISAID with the following filters: “low coverage excluded”, “collection date complete”, and “complete sequence”. Finally, we utilised these top hit sequences to compute each lineage’s frequency of mutations using a Python script (Pandas library version 1.4.2).

### 2.5. Network Analyses

To investigate the evolutionary history of the recombinants, we constructed a set that included the Brazilian recombinant, XD, and XS lineages and their respective putative parental sequences. However, we only added the XF sequences to the dataset since we did not identify recombination in this lineage with the Sc2rf analysis (i.e., there were no candidate parental sequences to be included).

Subsequently, we aligned the sequences using Nextalign and submitted the dataset to a network analysis in Splitstree version 4.18.2 [26]. For this purpose, we used the NeighborNet method and drew the network using the RootedEqualAngle method using the Wuhan/WH01/2019 (EPI_ISL_406798) sequence as the root.

We also carried out a network analysis using the library pegas 1.1 from R version 4.1.3 [27]. We randomly sampled five sequences per lineage (AY.101, AY.4, B.1.617.2, BA.1, BA.1.1, XD, XS) from the original aligned dataset to improve the resolution of the network. As before, we included all four sequences of the Brazilian recombinant in the sampled dataset. Finally, we determined the haplotypes using the function haplotype and carried out the network modelling using the haploNet method (default parameters).

### 2.6. Recombination Analyses

For recombination detection, we carried out two additional analyses. Firstly, we utilised the sampled dataset in the software RDP4 version 4.101 [28], using a “full exploratory recombination scan” (all methods with default parameters). Secondly, we performed an analysis with the HybridCheck R library version 1.0.1 [29]. For this analysis, we considered each segment’s oldest top hit to be the parental sequence.

Concerning the XF lineage, we used South Africa/NICD-N28358/2022 (Omicron) and South Africa/NHLS-UCT-GS-AF27/2021 (Delta) as the parental sequences, as described in Wang et al. (2022) [16]. Regarding the recombinant sequences described in this study, we annotated the genome mutations using the Coronapp [30]. We then drew the genome maps using the Python libraries, Seaborn and DNA features viewer 3.1.1 [31].

### 2.7. Phylogenetic Analyses

To investigate the phylogenetic history of the Brazilian recombinant segments, we concatenated the four identified sequences with their respective parental sequences (top hits of lineages AY.101 and BA.1.1). We then split the aligned sequences into two segments, considering the recombination breakpoint inferred in the HybridCheck analysis: the 5′ portion encompassed nucleotide positions 1–21,769 and the 3′ portion encompassed positions 21,770–29,903.

Next, we built a phylogenetic tree using IQ-Tree version 1.6.12 [32] with an automatically detected substitution model (option-m MFP) and 1000 ultrafast bootstrapping replicates. We then conducted a timetree inference and a “mugration” model using discrete PANGO lineages with Treetime version 0.8.6 [33]. Subsequently, we drew a chronogram tree using a script written in R ggtree library version 3.2.1 [34], colouring the branches according to the PANGO lineages.

### 2.8. Estimating the Age of Introgression

To estimate the date of the recombination event, we extracted the SNPs of the Brazilian recombinant with snp-sites version 2.5.1 [35], only outputting columns containing ACGT (option-c). We then calculated the coalescence time based on the formula described in Ward & Oosterhout (2015) [29], considering a mutation rate of 1.83 × 10^−6^ substitutions per site per day [36] and a genome size of 29,903 nucleotides, based on the reference genome (Wuhan/2019).

To evaluate the context of the co-circulating lineages in Brazil, we plotted a kernel density of the absolute frequencies of Brazilian sequences collected between June 2021 and June 2022 (assessed in GISAID on 29 July 2022). We generated the density plots considering the Brazilian regions with a script written in Python (Seaborn library), employing the kdeplot method with a smoothing parameter equal to 2 (bw_adjust = 2). Then, we assessed the association between the Brazilian regions (South and non-South) and Pango lineages (AY.101, BA.1.1, and other lineages) using the Chi-square test (SciPy version 1.8.1). *p*-values < 0.05 were considered statistically significant.

## 3. Results

### 3.1. Sampling, Data Acquisition, and Genome Assembly

The SARS-CoV-2 recombinant samples were independently identified and processed by each institution according to their routine sequencing testing. The clinical data available and the assembly metrics for the three sequenced genomes are summarised in Table A1.

### 3.2. Identification of the Brazilian Deltacron, AYBA-RS

Preliminary analyses assigned the genome sequence from Cruz Alta (Brazil/RS-FIOCRUZ-8390/2022) to the recombinant lineage XS. However, the first 20 kb of the genome presented a mutational pattern distinct from that of an XS archetype (Figure A1 in Appendix A).

Through the genomic surveillance routine of the State Rio Grande do Sul, we identified two more sequences similar to the Cruz Alta sequence; one from Porto Alegre (Brazil/SC2-9898/2022) and another from Santa Maria (Brazil/RS-315-66266-219/2022) (Table A1 and Figure A2). Additionally, we searched the GISAID database and found a sequence from Rio de Janeiro (Brazil/RJ-NVBS19517GENOV829190059793/2022) that was very similar to the recombinants of South Brazil.

Once we identified our sequences as putative recombinants, we detected possible recombination signals in their genomes with Sc2rf. This analysis indicated that the 5′ region (positions 1–21,845) came from a Delta lineage, and the 3′ region (positions 21,846–29,903) was from an Omicron lineage (Figure A3). Further analyses indicated that the 5′ genomic region resembled AY.101, and that the 3′ region was most similar to BA.1 or BA.1.1 (Table A2). Next, we built lineage-specific sequence databases and searched for the sequences most similar to each segment (5′ Delta and 3′ Omicron). We considered the oldest top-hit for each segment to be the parental sequences, and we compared their mutational signatures to those of the Brazilian recombinant sequences (Figure 1). In this analysis, all the Brazilian recombinant sequences presented similar patterns: their 5′ segment matched AY.101, and their 3′ region, the BA.1.1 lineage (Figure 1 and Figure A4). The substitution C10604T was found exclusively in all four sequences of the Brazilian recombinant (Figure A4). Since the recombinant found in this study does meet the requirements of the Pango nomenclature [37], we named it AYBA-RS, considering its parental lineages (AY.101 and BA.1.1) and the location of its origin (RS, Brazil).

### 3.3. Comparison between AYBA-RS and the Other Deltacrons

We compared the AYBA-RS sequences to those from other Deltacrons described in Cov-Lineages [17], namely XD, XF, and XS. Identification of the recombination blocks using HybridCheck [29] supported the above result with Sc2rf, revealing a breakpoint at the beginning of the gene S (Figure 2, position 21,769). Furthermore, the HybridCheck analysis showed that the recombination pattern differed from those of XD, XF, and XS (Figure 2). The XD and the AYBA-RS were mainly composed of a Delta scaffold, while the XF and XS were of an Omicron scaffold. Analysis with the RDP4 software [28] confirmed that the AYBA-RS arose from a single recombination event, separated from those that led to the other Deltacrons (Table A3). This analysis also indicated a breakpoint close to the gene S (position 22,675) (Table A3). The RDP4 analysis revealed recombination events for the XD and XS sequences but not for the XF sequences.

### 3.4. Evolutionary History of Recombinants of VOC Delta and VOC Omicron

The phylogenetic network reconstruction and haplotype network analysis were congruent since all four Brazilian recombinant sequences formed a group distinct from the other Deltacrons. Furthermore, both models showed that the Deltacrons were distributed between the Delta and Omicron groups, having additional portions of each lineage (Figure 3A,B).

Phylogenetic analyses for each 5′ (Delta) and 3′ (Omicron) block of the AYBA-RS assigned the 5′ segments of the Brazilian Deltacron to the AY.101 clade. This clade was formed only by sequences from Brazil, notably from Santa Catarina (SC), a state from the South region that borders the Rio Grande do Sul (RS) (Figure A5). On the other hand, the 3′ segments of the AYBA-RS formed a clade with BA.1.1 sequences from diverse geographical locations. However, in this tree, the AYBA-RS did not form a group with sufficient bootstrap support (Figure A5).

Considering the number of SNPs between the AYBA-RS sequences (Figure 2B, Table A4 and Table A5), we estimated that the recombination event that gave origin to the first AYBA-RS genotype was likely to have occurred 180 days before the collection date of the first sample, i.e., December 2021. Inspection of the lineage density plots revealed an overlap of AY.101 and BA.1.1, mainly in December 2021, across the country’s regions (Figure 4). In addition, AY.101 and BA.1.1 presented higher relative frequencies in the South region than in the rest of Brazil (Table A6, Chi-square test: X^2^ = 10,519.21, d.f. = 1, *p* < 0.00001).

## 4. Discussion

Here we describe the first Deltacron lineage identified in Brazil, AYBA-RS. Our analysis shows that this recombinant strain arose from a single recombination event between the AY.101 and the BA.1.1 lineages in Southern Brazil. The genetic exchange between both variants most likely happened in December 2021, when the Omicron lineage started to overtake Delta around the country [38]. Furthermore, we show that this recombinant differs from the previously described SARS-CoV-2 Deltacrons XD, XF, and XS, supporting a new recombination event and evidence of incipient parallel evolution. We employed a robust approach to identify and describe the recombinant SARS-CoV-2 lineages, combining methods involving phylogenetic and population genetic techniques incorporated in HybridCheck [29], RDP4 [28], SplitsTree [26] and other approaches. This combined approach enabled us to determine the parental lineages, identify the recombination breakpoint around the 22 kb position near the Spike gene (S), and estimate the date when the new recombinant lineage evolved.

In genomic studies of hybridisation, an apparent signature of genetic introgression can also be caused by mixed infections that result in chimeric sequences. Those chimeras may be erroneously interpreted as recombinants or hybrids. However, we observed four (nearly) identical recombinant genotypes that were collected at separate times and in different locations. Furthermore, these isolates were sequenced in other laboratories. Hence, we can confidently rule out the possibility of mixed infections resulting in chimeric sequences. Therefore, we can conclude that the samples described here are genuine recombinants.

Based on the phylogenetic reconstruction, we were able to ascertain that the sequences found in the Rio Grande do Sul and Rio de Janeiro States coalesced and had a single origin. Since genomic deposits in GISAID are recent, the absence of more sequences in the database suggests that the variant had a minor epidemiological impact. Alternatively, a lack of genomic surveillance may also have contributed. Indeed, the sequencing effort in Brazil is still lower than those in Europe and the USA [39], and this could have resulted in an underestimation of the true prevalence of this variant. The more comprehensive sequencing in Europe and the USA might explain why most of the recombinants are found in these regions. On the other hand, this could also reflect a genuine pattern, given that these regions experience considerable international air travel, enabling viral gene flow and between-variant recombination.

Co-infections with different variants of SARS-CoV-2 are necessary to trigger recombination events. Spatiotemporal variation in selection pressures can maintain a balanced polymorphism and multiple variants. In addition, multiple variants can also be maintained in substructured environments, as well as through a time lag in coevolution [40]. International travel can mediate gene flow and connect these distinct variants, facilitating inter-variant recombination and genetic introgression [3]. In fact, along the pandemic’s course, there have been reports of patients having Omicron and Delta co-infections [41,42,43]. Such events provide an opportunity for the emergence of new lineages with distinct phenotypes [2,44]. These phenotypes can occupy different peaks in the fitness landscape separated by fitness valleys [45,46]. Such valleys can be a consequence of epistasis, which is a phenomenon wherein nucleotide substitutions influence each other’s impact on fitness, resulting in a fitness landscape with many small and large peaks, ridges, and valleys. In such a rugged landscape, populations evolve slowly because they can become stuck once they have reached a local optimum, i.e., the highest fitness peak in the nearby landscape [47]. In that case, several mutations are required to climb the next even higher peak [48]. Recombination events could help the virus to bridge such valleys because recombination (and genetic introgression) offers three theoretical advantages over mutations (see Introduction). Given the large amount of nucleotide divergence that has evolved in multiple extant lineages, we argue that it is likely that recombinant evolution will play an increasingly important role in SARS-CoV-2 evolution and the COVID-19 pandemic. The potential for recombination to evolve better-adapted SARS-CoV-2 variants is increased by international travel that can bring allopatric lineages and variants from different continents together.

An analysis proposed by Turakhia and co-workers [49] suggested that approximately 2.7% of sequenced SARS-CoV-2 genomes have detectable recombinant ancestry. However, the authors also highlight that hybrid strains of genetically similar viral lineages are challenging to detect and that the overall recombination frequency could be underestimated [49,50]. The current study corroborates this assertion, showing that distinct recombinant lineages can be challenging to differentiate and that advanced evolutionary genomic analyses are required to identify and trace the origin of recombinant lineages. In addition, future studies with more Deltacron lineages would extend our analysis, allowing us to verify the breakpoint site’s impact on the recombinants’ fitness.

The genomic bulletin from June 2022, which included only 83 samples collected in the Rio Grande do Sul, revealed that even with the predominance of the Omicron lineage, Delta (AY.99.2) and Gamma (P.2) lineages are still circulating. Taking into account the relaxation of prevention measures, the non-adherence to the vaccine booster dose, and the simultaneous circulation of multiple lineages in the same region, we might be creating a perfect storm for the emergence of new SARS-CoV-2 VOCs. Our study supports the assertion that SARS-CoV-2 genetic introgression events might be more common than expected initially. This observation has implications for disease control measures, emphasising the need for more intensive genomic and epidemiological surveillance worldwide.

## Figures and Tables

**Figure 1 vaccines-11-00212-f001:**
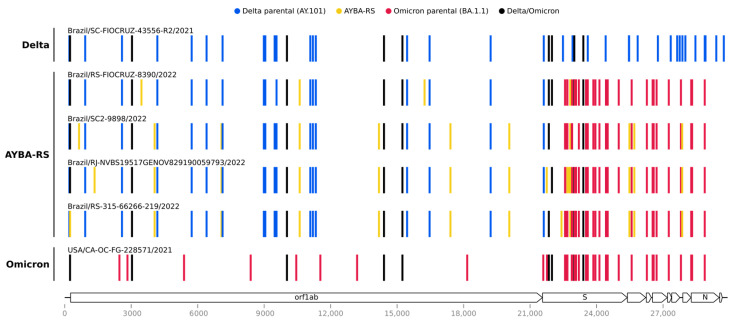
Mutation profiles of the AYBA-RS and the Delta and Omicron variants. The AYBA-RS recombinant presents a similar mutational pattern among the four Brazilian samples, where the first ~22 kb (5′ segment) resembles the Delta variant, and the last ~8 kb (3′ segment) the Omicron variant. Vertical colored lines represent nucleotide substitutions, with Wuhan/2019 as the reference. The legend depicts the characteristic mutations of each parental sequence and of the recombinants. The SARS-CoV-2 genome map and their respective coordinates are shown at the bottom.

**Figure 2 vaccines-11-00212-f002:**
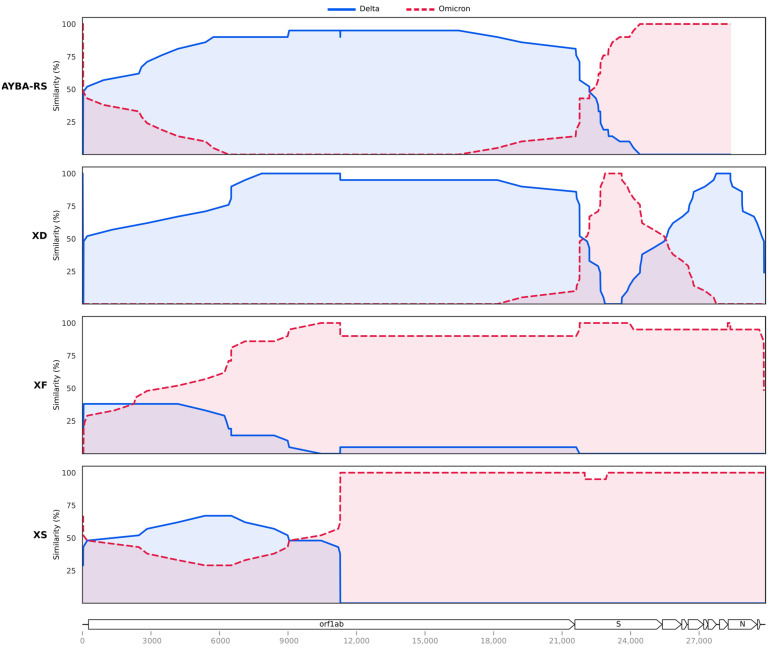
Comparison of the recombination patterns of the AYBA-RS Deltacron and the Deltacrons XD, XF, and XS. Similarity values between the recombinant and the Delta and Omicron parental sequences within a window length of 20 nucleotides along the genome. The legend describes the reference of each comparison. The recombination breakpoints, i.e., points where the lines cross over each other, are distinct for each of the recombinants, and this supports our conclusion of their independent evolution. Recombination plots generated with HybridCheck [29]. Sequences analysed in this plot: Brazil/RS-FIOCRUZ-8390/2022 (AYBA-RS), Brazil/SC-FIOCRUZ-43556-R2/2021 (Delta) and USA/CA-OC-FG-228571/2021 (Omicron) as its parental sequences; France/HDF-IPP54794/2022 (XD), Sweden/37524448XXP/2021 (Delta) and Finland/P-1301/2022 (Omicron) as its parental sequences; England/PHEC-YYN8J41/2022 (XF), South Africa/NHLS-UCT-GS-AF27/2021 (Delta) and South Africa/NICD-N28358/2022 (Omicron) as its parental sequences; USA/CO-CDC-FG-248528/2022 (XS), Latvia/3410639/2021 (Delta) and USA/CA-CDC-FG-223742/2021 (Omicron) as its parental sequences. Similarity at the y-axis refers to percentage sequence similarity at polymorphic sites only. The SARS-CoV-2 genome map and their respective coordinates are shown at the bottom.

**Figure 3 vaccines-11-00212-f003:**
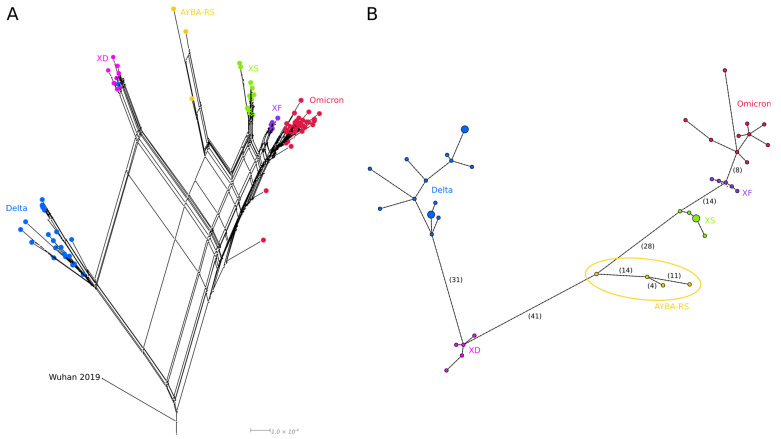
The evolution of Deltacron variants (XD, XF, XS and AYBA-S) in relation to that of their parental variants (Delta and Omicron). (**A**)—Phylogenetic network of Deltacron sequences built with SplitsTree (NeighborNet method, RootedEqualAngle on Wuhan/WH01/2019). The loops in this network are consistent with recombinant evolution. (**B**)—Haplotype network of Deltacron sequences made with pegas (haploNet method). The number of SNPs is in parentheses (Table A5); the AYBA-RS is within an ellipse. In both analyses, Delta (AY.101, AY.4, B.1.617.2) and Omicron (BA.1, BA.1.1) sequences were used as references. Nodes are coloured according to the variant type.

**Figure 4 vaccines-11-00212-f004:**
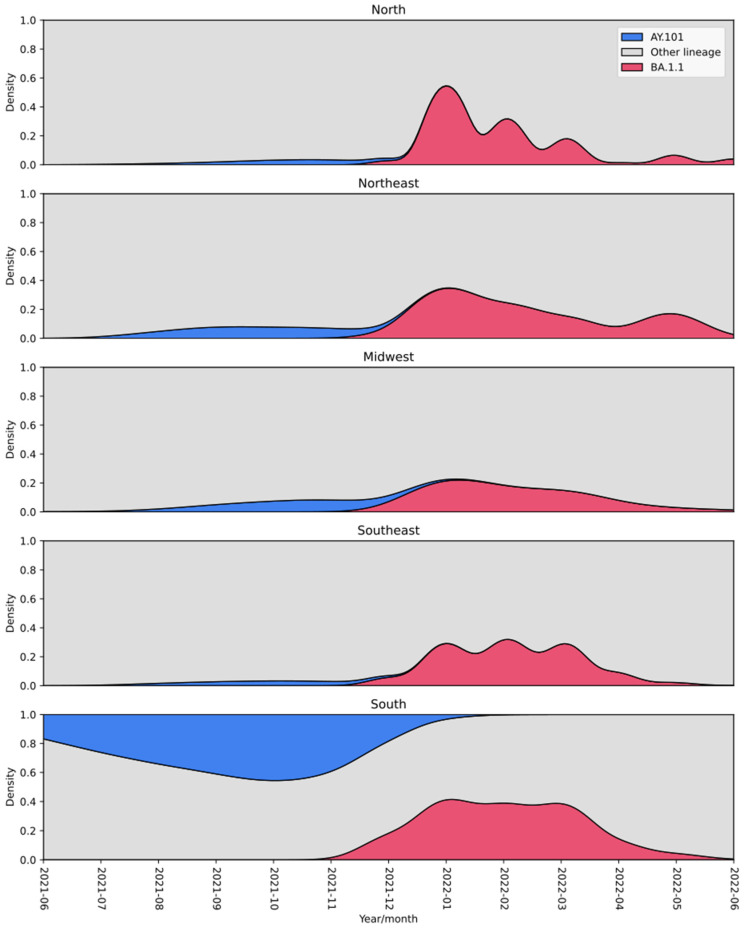
Density plot of the frequencies of AY.101 (Delta) and BA.1.1 (Omicron) lineages circulating in Brazil between June 2021 and June 2022. Both lineages co-circulated mainly in December 2021 in South Brazil. The colour of the lineages is shown in the legend.

## Data Availability

All genome sequences and associated metadata are published in GISAID’s EpiCoV database (EPI_SET_220829tz). To view the contributors of each sequence with details such as accession number, virus name, collection date, originating lab and submitting lab, and the list of authors, visit 10.55876/gis8.220829tz.

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
