# Peer review of "Incipient Parallel Evolution of SARS-CoV-2 Deltacron Variant in South Brazil"

_vaccines, 2023, doi:10.3390/vaccines11020212_

Round 1

Reviewer 1 Report

I have reviewed this article entitled, “Incipient parallel evolution of SARS-CoV-2 Deltacron variant in South Brazil.”

I have availed this opportunity to make a judgment and evaluate this exciting research paper. The topic of this research study has a practical significance to scientific knowledge. The authors have investigated a promising research area. I will recommend this paper for publication with some essential modifications. The authors have to revise this paper by following my recommendations. After making these minor changes as suggested below, I will accept this article. Modify according to these suggestions.

I am in favor of this paper for publication after the necessary changes. However, the authors must revise the manuscript and work according to my suggestions to enhance the quality. I will accept this paper for publication after these minor changes, as suggested below. Revise according to these suggestions. I will accept this interesting study after modifications.

Questionnaire / Appendix

Provide an Appendix at the end. Variables scales should be provided before references.

Abstract

In my opinion, I have some guidelines for the authors to enhance the study quality before endorsing it for publication. As the Abstract is the main door of the manuscript, it should briefly present high-quality English with new information.

Please revise the whole article and remove English grammar problems. I suggest the authors take English editing services from some agencies to improve the quality of this study.

Introduction section

I suggest that authors read and add the latest citations to the introduction, literature, and method sections to enhance the quality of the study.

Literature section:

Add literature section. You cannot delete this section. Read the suggested studies and cite these papers in the literature to enhance the quality of your work.

Su, Z., Cheshmehzangi, A., Bentley, B. L., McDonnell, D., Šegalo, S., Ahmad, J., . . . da Veiga, C. P. (2022). Technology-based interventions for health challenges older women face amid COVID-19: a systematic review protocol. Systematic Reviews, 11(1), 271. doi:10.1186/s13643-022-02150-9

Materials and Methods

This section needs improvement. Please follow the suggested studies and improve your paper. The authors need to improve this section. I am recommending some good studies. Read the methods of these studies, improve your writing, and cite these studies in this section. Suggested proper article citations.

Farzadfar, F., Naghavi, M., Sepanlou, S. G., Saeedi Moghaddam, S., Dangel, W. J., Davis Weaver, N., . . . Larijani, B. (2022). Health system performance in Iran: a systematic analysis for the Global Burden of Disease Study 2019. The Lancet. doi:10.1016/s0140-6736(21)02751-3

Discussion section:

Make a separate heading for the Discussion section. It should be around one page and a half. Make it strong. See the recommended studies and improve your sections.

Conclusion

Highpoint creativity and scientific contribution of this study to the body of literature. The English level needs corrections to meet scientific merit for publication. I accept and endorse this manuscript for publication after minor corrections, as suggested.

Reviewer 2 Report

Since the outbreak of COVID-19, Severe Acute Respiratory Syndrome Coronavirus 2 (SARS-CoV-2) has been capable of extending the pandemic by mutating into different variants. Genomic surveillance is critical for detecting and responding dynamically to new and changing SARS-CoV-2 variants, and it is the most effective tool for identifying emerging SARS-CoV-2 variants. Aside from mutation, recombination is a common evolutionary mechanism in coronaviruses that can result in a rapid accumulation of mutations.

In Brazil, the authors of this paper discovered the SARS-CoV-2 Deltacron recombinant variant. The 5' region (positions 1-21845) was found to be from a Delta lineage, while the 3' region (positions 21846-239 29903) was found to be from an Omicron lineage. The 5' genomic region was most similar to AY.101, and the 3' region was most similar to BA.1 or BA.1.1. This study provides new evidence of the recombination of SARS-CoV-2.

However, it would be more convincing if the authors could demonstrate that the sequencing result is from recombinant rather than a mix of Omicron and Delta coinfection.

There has been a report of Delta-Omicron recombinant in the US.(Lacek KA, et al. SARS-CoV-2 Delta-Omicron Recombinant Viruses, United States. Emerg Infect Dis. 2022 Jul;28(7):1442-1445.); it would be interesting if the author could compare the sequences of the two studies.
